# Stability and Detection Limit of Avian Influenza, Newcastle Disease Virus, and African Horse Sickness Virus on Flinders Technology Associates Card by Conventional Polymerase Chain Reaction

**DOI:** 10.3390/ani14081242

**Published:** 2024-04-21

**Authors:** Machimaporn Taesuji, Khate Rattanamas, Peter B. Yim, Sakchai Ruenphet

**Affiliations:** 1Clinic for Horse, Faculty of Veterinary Medicine, Mahanakorn University of Technology, Bangkok 10530, Thailand; machimapornt@gmail.com; 2Master of Science Program in Animal Biotechnology, Mahanakorn University of Technology, Bangkok 10530, Thailand; r.khate73@gmail.com (K.R.); peter@mut.ac.th (P.B.Y.); 3Immunology and Virology Department, Mahanakorn University of Technology, Bangkok 10530, Thailand

**Keywords:** African horse sickness virus, avian influenza virus, detection limit, flinders technology associates card, Newcastle disease virus, stability

## Abstract

**Simple Summary:**

The instability of viral RNA, which is susceptible to ubiquitous RNases, poses a significant challenge for its transport and detection in diagnosis. To address this challenge, this study aimed to evaluate the stability and detection limits of various RNA viruses, including the avian influenza virus, Newcastle disease virus, and African horse sickness virus, on Flinders Technology Associates cards. This investigation provides empirical evidence supporting the efficacy of Flinders Technology Associates cards for sample collection and subsequent viral RNA recovery, highlighting their suitability for use in molecular diagnostics. Consequently, based on the demonstrated effectiveness, stability, and safety implications observed in this study, Flinders Technology Associates cards are recommended for virus storage and transport, thus facilitating the molecular detection and identification of RNA viral pathogens.

**Abstract:**

The Flinders Technology Associates (FTA) card, a cotton-based cellulose membrane impregnated with a chaotropic agent, effectively inactivates infectious microorganisms, lyses cellular material, and fixes nucleic acid. The aim of this study is to assess the stability and detection limit of various RNA viruses, especially the avian influenza virus (AIV), Newcastle disease virus (NDV), and African horse sickness virus (AHSV), on the FTA card, which could significantly impact virus storage and transport practices. To achieve this, each virus dilution was inoculated onto an FTA card and stored at room temperature in plastic bags for durations ranging from 1 week to 6 months. Following storage, the target genome was detected using conventional reverse transcription polymerase chain reaction. The present study demonstrated that the detection limit of AIV ranged from 1.17 to 6.17 EID_50_ values over durations ranging from 1 week to 5 months, while for NDV, it ranged from 2.83 to 5.83 ELD_50_ over the same duration. Additionally, the detection limit of AHSV was determined as 4.01 PFU for both 1 and 2 weeks, respectively. Based on the demonstrated effectiveness, stability, and safety implications observed in the study, FTA cards are recommended for virus storage and transport, thus facilitating the molecular detection and identification of RNA viral pathogens.

## 1. Introduction

Avian influenza viruses (AIVs) pose substantial obstacles to worldwide public health systems due to their extensive spread and notable mortality rates [1]. Falling under the influenza A genus, AIVs possess an eight-segment genome and code for a minimum of 11 distinct proteins, notably hemagglutinin (HA) and neuraminidase (NA) glycoproteins. Avian species categorize HA into sixteen subtypes and NA into nine subtypes, respectively, and these proteins subdivide to ascertain distinctive serotypes of AIV based on their genetic differences [2,3]. Furthermore, based on their pathogenicity to chickens, AIVs are further classified into two groups determined by the intravenous pathogenicity index (IVPI) test: highly pathogenic avian influenza viruses (HPAIVs) and low-pathogenic avian influenza viruses (LPAIVs) [4,5]. Recent transmissions of HPAIV strains like H5N1, H5N8, and H7N9 have presented considerable risks to public health [6]. With H5N1 being acknowledged as the most virulent within this group, it has demonstrated elevated mortality rates in both avian and human populations [1]. The first instance of the HPAI H5N1 virus affecting poultry in Scotland in 1959, and subsequently transmitting to humans in Hong Kong in 1997, marked the beginning of several documented outbreaks of HPAIV. These outbreaks have been evidenced by various researchers [1,2,4,7]. During the period spanning 2003 to 2023, the World Health Organization (WHO) registered 868 cases of HPAI H5N1 infection in humans, with 457 fatalities (52.64%) occurring across 21 countries [8]. Additionally, Hinjoy et al. [9] reported that Thailand has been free of reported cases of avian influenza since 2008. Nonetheless, AIVs circulating in poultry in neighboring countries could still pose a risk of transmission to humans.

Several of the most devastating diseases from animals are caused by pathogens that exhibit a broad host range, affecting both mammals and birds. For example, avian viruses of the subfamily Avulavirinae with 22 serotypes [10] are known to cause such diseases. Over 200 species of wild birds spanning 27 orders act as the primary reservoir for Avulavirinae, underscoring the extensive diversity involved. The migratory patterns of these avian species play a pivotal role in the dissemination of the pathogen, fostering transmission within and among species. Notably, the Newcastle disease viruses (NDVs) within Avulavirinae stand out as significant, having triggered five panzootics, with the latest one still underway [11]. Newcastle disease, characterized by its high contagion rate, exacts substantial economic tolls and presents a severe menace to the worldwide poultry sector. Unregulated commerce in live birds and associated markets emerges as a pivotal conduit for the propagation of hazardous viruses, encompassing NDV and HPAI viruses [11,12,13,14].

Across numerous centuries, the relentless African horse sickness (AHS) has persistently tormented horse owners throughout sub-Saharan Africa. Though infectious yet non-contagious, this illness results in elevated mortality rates among vulnerable hosts, exacerbating the challenges faced by horse owners in sub-Saharan Africa. Consequently, due to its severity and the potential threat it poses for swift worldwide dissemination, AHS is categorized as a notifiable viral ailment by the World Organization for Animal Health (WOAH, formerly known as OIE) [15]. AHS retains its position as the most economically consequential equine sickness globally, emphasizing its profound influence on the equine industry. It has been established that the illness is attributed to nine separate serotypes of the AHS virus (AHSV), distinguished by a complex linear double-stranded RNA genome belonging to the Orbivirus genus within the family Reoviridae [16,17,18]. The notion of AHSV transmission via blood-feeding arthropods was initially proposed by Pitchford and Theiler in 1903, further elucidating the mechanisms underlying the spread of AHS [19]. Several studies demonstrated the infection of mixed pools of wild-caught Culicoides species with AHSV and observed AHSV replication within Culicoides species after oral ingestion [20,21,22]. Recent outbreaks, such as the confirmation in Thailand during March 2020, highlight the ongoing threat of global AHS outbreaks. In light of recent outbreaks, researchers determined the causative pathogen of AHS to be serotype 1, closely related to isolates from South Africa [22,23,24,25].

Generally, AIV, NDV, and AHSV require the demonstration of viral antigens or nucleic acid for diagnosing the cause of disease. However, the characteristic clinical signs are not specific and could be confused with those of other diseases. Therefore, appropriate sample collection, sampling techniques, and safety measures are crucial for confirming cases of the disease while minimizing the risk of disease transmission. Furthermore, it is well known that the detection and characterization of viral RNA pathogens in fieldwork are challenging due to the instability of the RNA molecule. Despite these challenges, several researchers have mentioned the benefit of using the Flinders Technology Associates (FTA) card for sample storage to identify important viral diseases in animals. However, optimal handling, processing, and biosafety measures are not well established [26,27,28,29].

The FTA paper, pioneered by Lee Burgoyne at Flinders University Australia, represents a groundbreaking approach to DNA storage, complemented by patented cards from Whatman International Ltd. and distributed by Flinders Technology Associates, providing an efficient solution for ambient temperature collection, transportation, and long-term storage of DNA [30]. The FTA card, comprising a cotton-based cellulose membrane infused with a chaotropic agent, assumes a critical function in safeguarding sample integrity and safety through the inactivation of pathogens, cellular lysis, and the preservation of nucleic acid within the fiber matrix [31]. As a result, these samples are rendered non-infectious, thereby eliminating any biohazard potential [32]. Nonetheless, the efficacy of FTA cards in detecting RNA viruses in animals, particularly RNA viruses, remains ambiguous, highlighting a crucial area warranting further investigation. This study seeks to bridge this lacuna by systematically assessing the stability and detection limit of diverse RNA viruses, including AIV, NDV, and AHSV, on FTA cards. This evaluation employed conventional reverse transcription polymerase chain reaction (RT-PCR) techniques for precise nucleic acid detection.

## 2. Materials and Methods

### 2.1. Candidate

The Flinders Technology Associates (FTA) card (FTA^TM^ Classic Card, GE Healthcare, Buckinghamshire, UK), bearing lot number 17008362, was utilized for the present study.

### 2.2. Virus Sample Preparation

The low-pathogenic avian influenza virus (AIV), namely A/Duck/Asiancountry/2004 H9N2, and high-virulent Newcastle disease virus (NDV), namely NDV/chicken/Aseancountry/2013 [33], used in this study were courtesy of Prof. Dr. Thaweesak Songserm (Department of Pathology, Faculty of Veterinary Medicine, Kasetsart University). These viruses were propagated in 9-day-old chicken embryonic eggs. After allantoic fluid harvesting at 3 days post-inoculation, stock viruses were aliquoted and kept at −80 °C until testing. Both viruses were titrated using chicken embryonic eggs, following the methods described by Ruenphet et al. [34]. Additionally, the live attenuated African horse sickness vaccine serotype 1 (Onderstepoort biological product, Pretoria, South Africa) was dissolved following the manufacturer’s recommendation. The dissolved vaccine virus was aliquoted for stocking and kept at −80 °C until testing.

### 2.3. AIV Inactivation on FTA Card

To compare the virus-inactivating efficiency of FTA cards with regular filter paper (Whatman™, GE Healthcare, Buckinghamshire, UK), AIV was dropped onto both types of paper, with a volume of 100 μL per sheet for 4 sheets. Following this, the paper was allowed to dry in a level 2 biosafety cabinet for 30 min. Subsequently, samples were collected at specific time intervals: 30 min, 1 day, 3 days, and 7 days. Then, paper samples from both groups were tested for the viability of the AIV. Firstly, the whole sheet of paper containing the virus was cut into small pieces and placed in a 1.5 mL microtube containing phosphate-buffered saline (PBS). The samples in the microtube were then mixed well with a vortex and centrifuged with a temperature-controlled centrifuge at 4 °C at 10,000× *g* for 3 min. The resulting centrifuged liquid was injected into 9-day-old chick embryonic eggs to evaluate the virus’s viability on both types of papers. After incubating the inoculated embryonic eggs for 4 days, the allantoic fluid was tested for viability and/or for the increase in the number of AIVs using a hemagglutination (HA) test. Allantoic fluid was collected and reinjected into the embryonic eggs for samples that gave negative results. Following an additional 4 days of incubation, the allantoic fluid of embryonic eggs was tested again for HA. A negative result at this stage indicates the absence of live virus in either paper sample. However, a positive result indicates that both papers cannot effectively inactivate the virus.

### 2.4. Stability and Detection Limit Determination

Each virus was diluted as a 10-fold serial dilution using PBS before being inoculated onto the FTA card, in order to facilitate subsequent analysis. Subsequently, each dilution of every virus was inoculated with 100 μL onto the FTA card. Following inoculation, the cards were placed in a class II safety cabinet for 30 min to inactivate the virus. Subsequently, the inoculated FTA cards were stored at room temperature in a plastic airtight container for durations ranging from 1 week to 6 months. Following storage, the infected FTA card was punctured using a 2.5 mm diameter biopsy dermal puncher for AIV and NDV, and an 8.0 mm diameter for AHSV. After puncturing, RNA isolation for AIV and NDV was performed on the FTA card using an FTA purification reagent (Whatman^TM^, Maidstone, Kent, UK) according to the manufacturer’s recommendation. Subsequently, the punctured FTA card was placed into a microtube, and 180 μL of the purification reagent was added, with incubation at room temperature lasting for 5 min. Following incubation, the punctured FTA card was pipetted using the same purification reagent in the microtube 15 times, ensuring through mixing and extraction. This process was repeated three times for thorough RNA isolation. Following RNA isolation, 180 μL of TE buffer (10 mM Tris–HCl, 0.1 mM EDTA, pH8.0) was added, and the mixture was incubated at room temperature for 5 min. Subsequently, pipetting using the same TE buffer in the microtube was performed 15 times, followed by discarding all liquid to prepare for subsequent analysis. Lastly, the microtube containing the punctured card was incubated in a 50 °C incubator for drying for at least 15 min.

Following isolation on a punctured card, RNA from AIV was amplified by one-step reverse transcription polymerase chain reaction (RT-PCR) using the Superscript^TM^ III one-step RT-PCR system with Platinum^TM^ Taq DNA polymerase (Invitrogen^TM^, Life Technologies, Carlsbad, CA, USA). The RT-PCR reaction mixture consisted of 12.5 μL of 2× reaction mix, 0.5 μL of 10 pmol/μL of forward primer, 0.5 μL of 10 pmol/μL of reverse primer, 1 μL of Superscrip^®^ III RT/platinum^®^ taq mix, and 5.5 μL of double distilled water. The genomic viral RNA was amplified using a thermal cycler under the following condition: RT at 55 °C for 30 min, pre-denaturation at 95 °C for 5 min, PCR for 40 cycles containing denaturation at 94 °C for 1 min, annealing at 55 °C for 1 min, extension at 72 °C for 1 min, and post-extension at 72 °C for 10 min.

While the isolated card of infected NDV was also subjected to the Superscript^TM^ III One-Step RT-PCR System with Platinum^TM^ Taq DNA Polymerase, the thermal conditions were as follows: RT at 50 °C for 30 min, pre-denaturation at 94 °C for 15 min, PCR for 35 cycles containing denaturation at 94 °C for 30 s, annealing at 55 °C for 1 min, extension at 68 °C for 1 min, and finally post-extension at 68 °C for 7 min.

Furthermore, the AHSV-infected card was punctured at 8.0 mm diameter using TE buffer with a commercial extraction kit. Briefly, after the punctured FTA card was mixed with 250 μL of TE buffer and incubated at room temperature for 10 min, the entire liquid was transferred to a new microtube for genomic extraction using an extraction kit (GF-1viral nucleic acid extraction kit, Vivantis Technologies Sdn Bhd, Selangor Darul Ehsan, Malaysia). Following extraction according to the manufacturer’s recommendation, the extracted AHSV was amplified by two-step RT-PCR using a Viva 2-Step RT-PCR kit with M-MuLV RT/Taq DNA Polymerase (Vivantis Technologies Sdn Bhd, Malaysia). Briefly, the RNA reverse primer mixture comprised 1 μL of reverse primer mixed with 1 μL of dNTP mixture and 8 μL of extracted RNA, which was then incubated at 65 °C for 5 min and immediately kept on ice for 2 min. The RT master mixture consisted of 2 μL of 10× buffer M-MULV, 1 μL of M-MULV reverse transcriptase, and 7 μL of double distilled water. The RNA reverse primers and the RT master mix were then combined for complementary DNA synthesis using a thermal cycler with thermal conditions comprising of 42 °C for 60 min, 85 °C for 5 min, and 12 °C for 65 min for reverse transcription. Subsequently, complementary DNA were added to the PCR master mix, which comprised 2 μL of 5× pulsion buffer, 0.1 μL of pulsion polymerase, 0.2 μL of 10 mM dNTP mixture, 0.5 μL of forward primer, 0.5 μL of reverse primer, and 5.7 μL of double distilled water. The mixture underwent thermal cycling under the following conditions: pre-denaturation at 94 °C for 5 min, PCR for 35 cycles containing denaturation at 94 °C for 30 s, annealing at 53 °C for 30 s, and extension at 72 °C for 30 s, followed by post-extension at 72 °C for 10 min.

In the present study, we provided detailed descriptions of all primers used, as listed in Table 1. The PCR product was detected by gel electrophoresis, revealing specific amplicon sizes corresponding to AIV, NDV, and AHSV at 245, 305, and 1167 bp, respectively. These findings are crucial for confirming the presence of these viruses and assessing their genomic characteristics within the context of our study.

## 3. Results

### 3.1. AIV Inactivation on FTA Card

To compare the virus-inactivating efficiency of the FTA card with regular filter paper, the test results are as follows: no viability of the AIV was found on the FTA card from 30 min onwards. Additionally, the avian influenza virus’ viability was detected on the filter paper for up to 2 days but not beyond 3 days, as shown in Table 2. These results indicated that the FTA card could inactivate AIV within 30 min and could not infect in host cells, especially chicken embryonic eggs. On the other hand, the virus detected on the filter paper could infect embryonic eggs for up to 2 days.

### 3.2. Tested Titer Calculation

The present study aimed to calculate the AIV or NDV titer from three sources: the stock virus, the virus titer on the FTA card, and the tested titer, all obtained from FTA card samples used for genomic detection by 50% egg infection dose (EID_50_). Subsequently, the ratio of virus titer on the FTA card (diameter 25 mm) to the tested titer (diameter 2.5 mm) was calculated for each dilution. For instance, 100 μL of undiluted AIV resulted in a virus titer on the FTA card of 8.17 log_10_ EID_50_. However, a complication arose concerning the sampling area on the infected FTA card for genomic isolation and RT-PCR detection, which was at a ratio of 1:100 (Figure 1). Therefore, the tested titer of the undiluted virus was determined to be 6.17 log_10_ EID_50_. Furthermore, AHSV used a sampling diameter of 8.0 mm out of the 25 mm diameter of the FTA card, resulting in a ratio of 1:9.76 (Figure 1). Hence, the virus titer of the stock virus, virus titer on the FTA card, and the tested titers of undiluted AHSV were 6.00, 5.00, and 4.01 log_10_ plaque-forming unit (PFU), respectively.

### 3.3. Stability and Detection Limit

The present study aimed to calculate the AIV titer of the stock virus, the virus titer on the FTA card, and the tested titer for virus detection, yielding values of 9.17, 8.17, and 6.17 log_10_ EID_50_ of the undiluted virus (0 log_10_), respectively. This calculation was crucial for evaluating the effectiveness of virus detection methods. Additionally, the study determined the detection limit results over various durations, ranging from 1 and 2 weeks to 1, 2, 3, 4, and 5 months, yielding values of 1.17, 3.17, 4.17, 4.17, 5.17, 5.17, and 6.17 EID_50_, respectively (Figure 2 and Table 3). Subsequently, the study investigated the NDV titers of the stock virus, on the FTA card, and the tested titer, which were found to be 9.83, 8.83, and 6.83 log_10_ 50% embryonic egg lethal dose (ELD_50_) of the undiluted virus, respectively. The determination of NDV titers provided essential insights into the comparative effectiveness of different sample sources. Furthermore, the study analyzed the detection limit results for NDV, testing durations from 1 to 2 weeks up to 1, 2, 3, 4, and 5 months, resulting in values of 2.83, 3.83, 3.83, 3.83, 3.83, 5.83, and 5.83 ELD_50_, respectively (Figure 3 and Table 4). Finally, the detection limit of AHSV was determined as 4.01 PFU for both 1 and 2 weeks, indicating the sensitivity of the detection method (Figure 4 and Table 5). This comprehensive analysis of virus titers and detection limits enhances our understanding of virus detection methods and their applicability in various contexts.

## 4. Discussion

The utilization of FTA cards is widespread in the preservation and analysis of DNA, notably in forensic investigations and genetic research [38,39]. These cards are composed of filter paper saturated with a patented chemical mixture containing denaturants and a scavenger for free radicals. The chemical blend in FTA cards disrupts cells and organelles, inhibits bacterial proliferation, and alters protein structures, thus aiding in the capture and preservation of DNA. Consequently, the DNA remains firmly attached to the FTA cards, while proteins and inhibitors are rinsed off, ensuring stable long-term storage at room temperature [40]. This approach holds appeal for diverse applications, including studies in bacterial genetics, where maintaining DNA stability at room temperature is essential [41]. Moreover, FTA cards have been employed for sample storage and molecular detection of both DNA and RNA viruses, highlighting their versatility in various research fields. In recent times, proposals have advocated for the utilization of FTA cards as a replacement for liquid media in cervical cancer screening to detect human papillomavirus [42]. Additionally, improved biosafety has been cited as an advantage in storing and transporting specimens containing AIV [43]. However, the instability of viral RNA has been a significant challenge due to its susceptibility to ubiquitous RNases, complicating both its transport and detection. Therefore, this study aimed to evaluate the stability and detection limits of various RNA viruses, especially AIV, NDV, and AHSV, on FTA cards based on nucleic acid detection using conventional RT-PCR. This evaluation was conducted in response to the challenge posed by the instability of viral RNA in detection assays.

In the current investigation, we examined the stability of the FTA card at room temperature, focusing on its durability for a minimum of 5 months for AIV and NDV, and 2 weeks for AHSV, respectively. This exploration is significant for understanding the practical application of the FTA card in virus detection and storage. Concerning RNA storage, our study demonstrated significantly prolonged storage capability compared to the findings of Wannaratana et al. [29], who reported a preservation duration of at least 30 days for viral DNA on the Whatman™ FTA filter paper card. Moreover, Awad et al.’s [44] investigation revealed that various RNA viruses could be detected in the genomic of metapneumovirus on the FTA card when stored at 4–6 °C for up to 60 days. Additionally, Krambrich et al. [45] observed the detection of the Sindbis, Chikungunya, and Japanese encephalitis viruses, which remained detectable when stored in the infected FTA card at 4 °C and 25 °C over a 30-day period but degraded rapidly at 37 °C. However, our study discerned an inverse correlation between the detection limit and storage duration, particularly pronounced for AIV and NDV, as detailed in Table 3 and Table 4. Furthermore, the detection limit in our study accurately paralleled the actual virus titer on the FTA card and could also depict the detected titer, aligning with the reported findings of Rattanamas et al. [46]. This alignment with previous research emphasizes the reliability of our study’s findings and underscores the practical implications for virus detection and storage strategies.

Avian influenza viruses (AIVs) demonstrate a broad host tropism, infecting a diverse array of host species, encompassing both domesticated avian species and various wild avifauna. Certain highly pathogenic avian influenza viruses (HPAIVs) have been documented to induce severe and often fatal illness in humans [47]. Achieving a precise laboratory diagnosis of AIV infections demands meticulous adherence to time-consuming and logistically intricate precautionary measures during specimen or virus shipment, in order to mitigate potential biohazard exposure risking its effectiveness. Abdelwhab et al. [31] conducted a comprehensive investigation illustrating the effectiveness of utilizing FTA cards in preserving AIV infectivity and viral RNA’s integrity for subsequent detection, through various techniques including real-time RT-PCR, sequencing, and DNA microarray assay. The study revealed complete inactivation of infectivity for AIV subtype H6N2 and HPAIV subtype H5N1 within a mere one-hour period, following adsorption onto the FTA card under ambient room temperature conditions. The stability assessment indicated that viral RNA retained its integrity on FTA cards for a duration spanning 5 months. Conversely, our investigation showcased the ability to achieve AIV inactivation within a considerably reduced timeframe of 30 min, while also successfully detecting AIV RNA on the FTA card for an extended period of 5 months, thus suggesting notable progress beyond prior scholarly endeavors in this domain. Notably, FTA cards emerged as a viable option for analyzing field samples, albeit with a discernible reduction in AIV RNA detection sensitivity in comparison to the conventional method of direct swab examination, as observed in the study conducted by Abdelwhab et al. [31]. Employing FTA cards streamlines the secure transportation of samples intended for molecular AIV diagnosis, obviating the necessity for continuous cold storage, thereby amplifying both diagnostic efficacy and adherence to safety protocols.

Perozo et al. [48] conducted a study demonstrating the efficacy of FTA cards in detecting NDV RNA using allantoic fluid and chicken tissue samples. Their methodology involved Trizol^®^ RNA extraction and one-step RT-PCR. FTA cards play a pivotal role in facilitating the identification of NDV. Perozo et al. observed that NDV RNA extracted from allantoic fluid, boasting a titer of 5.8 log_10_ ELD_50_, exhibited stability on the cards for a span of 15 days. In contrast, our study detected varying levels of NDV RNA, with 2.83 log_10_ ELD_50_ detected after the infected card was kept at room temperature for 1 week, increasing to 5.83 log_10_ ELD_50_ for up to 5 months. Additionally, our study confirmed virus inactivation on the card within 30 min, consistent with reports by Rattanamas et al. [46] and Perozo et al. [48]. FTA cards unequivocally prove themselves to be suitable for collecting and transporting NDV-positive samples, providing a reliable source of RNA for molecular characterization, and ensuring sample safety.

AHS was identified as a newly emerging disease following an outbreak in northeastern Thailand in March 2020 [23,24,25,49]. King et al. [24] documented the inaugural AHS outbreak in Southeast Asia, providing crucial insights into its emergence following the outbreak in northeastern Thailand in March 2020. The researchers collected and analyzed four EDTA blood samples and four serum samples from horses in the Nakorn Ratchasima province of Thailand. Subsequently, these samples were sent to the Pirbright Institute, United Kingdom, through a private veterinarian for further analysis. Nucleic acid extraction was conducted on EDTA blood samples, followed by real-time RT-PCR and conventional RT-PCR techniques to precisely determine the AHSV serotype. The presence of AHSV serotype 1 was confirmed [24,50]. This study builds upon the methodologies established by King et al. [24], and conclusively demonstrates the effectiveness of utilizing FTA cards for safe sample transportation, facilitating molecular AHSV diagnosis. This method eliminates the requirement for constant cold storage, which is particularly noteworthy as AHSV can remain viable on FTA cards at room temperature for a minimum of 2 weeks, consequently enhancing both diagnostic accuracy and adherence to safety protocols. However, the AHSV amplicon size of 1167bp, compared with the amplicon sizes for AIV and NDV (of 245bp and 305bp, respectively) is much shorter. Therefore, longer amplifications (such as 1167 bp for AHSV) may experience lower sensitivity, possibly due to more breaks in the whole genome during the prolonged duration. Importantly, this study represents the first report of AHSV detection using the FTA card.

## 5. Conclusions

The current investigation provides empirical evidence supporting the efficacy of FTA cards for sample collection and subsequent recovery of viral RNA, both critical stages in downstream analysis due to their potential to simplify and enhance these processes. The established efficacy of FTA cards in sample collection and RNA recovery, as demonstrated in this investigation, renders them highly desirable for fieldwork dedicated to molecular detection and the characterization of viral pathogens. This is particularly pertinent in contexts favoring straightforward and economical sample collection and transport methodologies, where the effectiveness and practicality of FTA cards shine. The stability and detection limit of RNA genomic material, particularly evident for AIV, NDV, and AHSV as elucidated in the present study, serve to reinforce the benefits associated with the utilization of FTA cards for fieldwork concerned with molecular detection and the characterization of viral pathogens. Despite the potential inactivation of numerous viruses during collection and storage procedures, our study underscores the imperative of treating all specimens as potentially infectious. Particularly noteworthy is the retention of viable viruses for at least 2 days on regular filter paper, emphasizing the necessity for meticulous handling and storage protocols, which is relevant to the broader discussion on sample collection and storage methods. Therefore, considering the demonstrated effectiveness, stability, and safety implications, FTA cards emerge as a favorable choice for virus storage and transport in the molecular detection and identification of RNA viral pathogens, offering a reliable solution for fieldwork applications. Nonetheless, despite this study’s promising findings, the necessity for additional field testing to validate and standardize these methods for practical fieldwork applications remains imperative to ensure their widespread and effective use.

## Figures and Tables

**Figure 1 animals-14-01242-f001:**
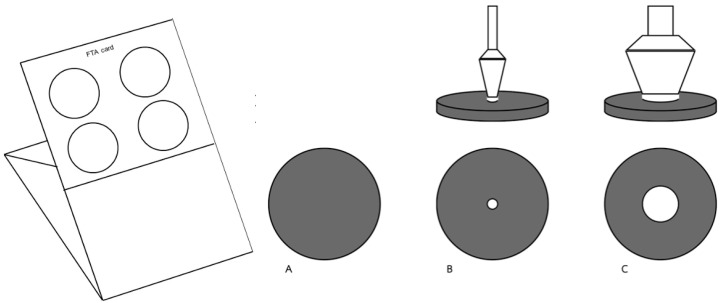
Illustrates the tested titer ratio, providing insight into the relationship between the virus titer on the Flinders Technology Associates (FTA) card and sampling for RNA detection using a biopsy dermal puncher. The illustration includes (**A**) the virus inoculation area (diameter 25 mm) = 491.07 mm^2^, (**B**) the sampling area for a 2.5 mm puncture = 4.91 mm^2^, with a ratio of B: A = 1:100, and (**C**) the sampling area for an 8.0 mm puncture = 50.28 mm^2^, with a ratio of C: A = 1:9.76. These ratios help in understanding the proportion between the different sampling areas and the virus titer on the FTA card, considering both the virus titer on the sampling FTA card (white circle area in (**B**,**C**)) and the virus titer in the inoculation area (dark circle area in (**A**)).

**Figure 2 animals-14-01242-f002:**
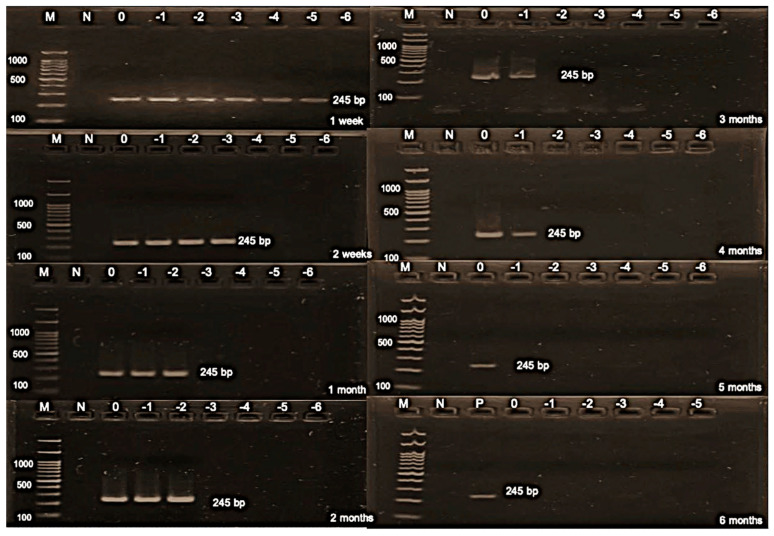
Gel electrophoresis of stability and detection limit of the avian influenza virus on the Flinders Technology Associates (FTA) card. M: DNA ladder; N: negative control; P: positive control; 0: undiluted; −1: diluted at 1 log_10_; −2: diluted at 2 log_10_; −3: diluted at 3 log_10_; −4: diluted at 4 log_10_; −5: diluted at 5 log_10_; and −6: diluted at 6 log_10._

**Figure 3 animals-14-01242-f003:**
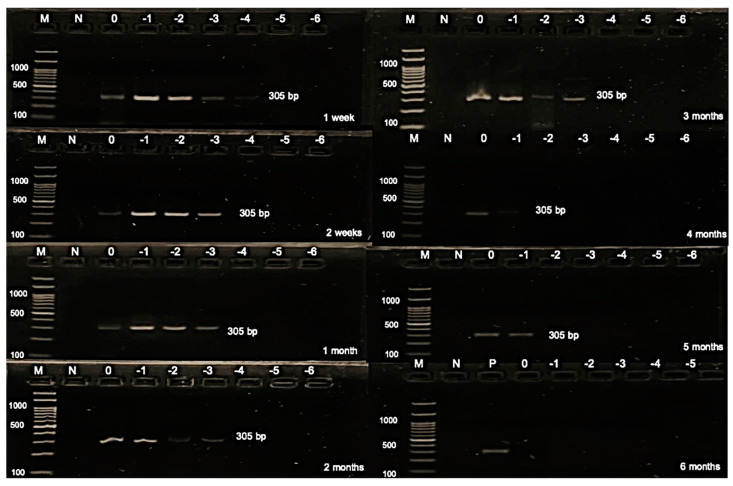
Gel electrophoresis of stability and detection limit of the Newcastle disease virus on the Flinders Technology Associates (FTA) card. M: DNA ladder; N: negative control; P: positive control; 0: undiluted; −1: diluted at 1 log_10_; −2: diluted at 2 log_10_; −3: diluted at 3 log_10_; −4: diluted at 4 log_10_; −5: diluted at 5 log_10_; and −6: diluted at 6 log_10._

**Figure 4 animals-14-01242-f004:**
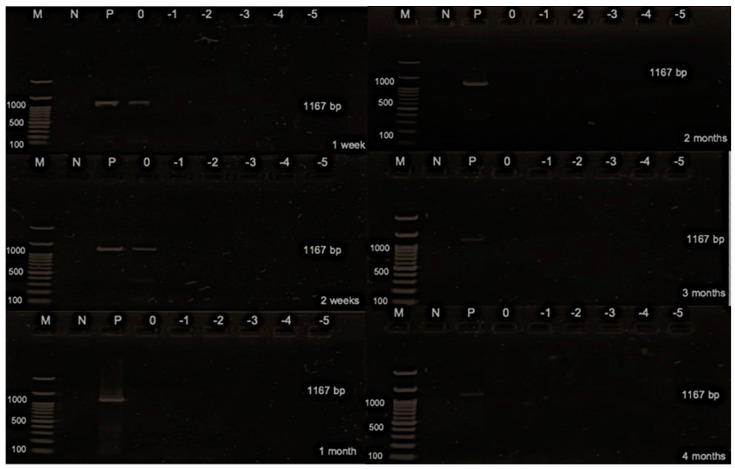
Gel electrophoresis of stability and detection limit of the African horse sickness virus on the Flinders Technology Associates (FTA) card. M: DNA ladder; N: negative control; P: positive control; 0: undiluted; −1: diluted at 1 log_10_; −2: diluted at 2 log_10_; −3: diluted at 3 log_10_; −4: diluted at 4 log_10_; −5: diluted at 5 log_10_; and −6: diluted at 6 log_10._

**Table 1 animals-14-01242-t001:** List and sequences of primers designed for the avian influenza virus (AIV), Newcastle disease virus (NDV), and African horse sickness virus, which were utilized for experimentation in this study.

Virus	Primer Name	Primer Sequence (5′ to 3′)	Target Gene	Amplicon Size (bp)	References
AIV	FluA-M52C_F	CTTCTAACCGAGGTCGAAACG	Matrix	245	Fouchier et al., 2000 [35]
	FluA-M253_R	AGGGCATTTTGGACAAAKCGTCTA			
NDV	NOH_F	TACACCTCATCCCAGACAGG	Fusion	305	Capua and Alexader, 2009 [36]
	NOH_R	AGTCGGAGGATGTTGGCAGC			
AHSV	AHSV_F7	GTTAAAATTCGGTTAGGATG	Segment 7	1167	Zientara et al., 1994 [37]
	AHSV_R7	GTAAGTGTATTCGGTATTG			

**Table 2 animals-14-01242-t002:** Avian influenza virus’ inactivation on the Flinders Technology Associates (FTA) card compared with regular filter paper using chick embryonic eggs.

Time	1 dpi		2 dpi		3 dpi		4 dpi		HA Test	
	FTA	Filter Paper	FTA	Filter Paper	FTA	Filter Paper	FTA	Filter Paper	FTA	Filter Paper
30 min	− − − ^a^	− − −	− − −	+ + + ^b^	− − −	+ + +	− − −	+ + +	− − − ^c^	+ + + ^d^
1 day	+ ^b^ − −	− − −	+ − −	− − −	+ − −	− − −	+ − −	− − −	− − − ^c^	+ + + ^d^
3 days	− − v	− − −	− − −	+ ^b^ − −	− − −	+ − −	− − −	+ − −	− − − ^c^	− − − ^c^
7 days	− − −	+ − −	− − −	+ ^b^ − −	− − −	+ − −	− − −	+ − −	− − − ^c^	− − − ^c^

dpi: day post-inoculation; ^a^ alive embryo; ^b^ dead embryo; ^c^ negative agglutination; and ^d^ positive agglutination.

**Table 3 animals-14-01242-t003:** Stability and detection limit of the avian influenza virus using stock virus comparison on Flinders Technology Associates (FTA), using conventional reverse transcription polymerase chain reaction (RT-PCR).

Dilution (log_10_)	Virus Titer (log_10_ EID_50_/_mL_)	Virus Titer on FTA (log_10_ EID_50_)	Tested Titer (log_10_ EID_50_)	Stability on FTA							
				1 wk	2 wk	1 m	2 m	3 m	4 m	5 m	6 m
0	9.17	8.17	6.17	+	+	+	+	+	+	+	−
1	8.17	7.17	5.17	+	+	+	+	+	+	−	−
2	7.17	6.17	4.17	+	+	+	+	−	−	−	−
3	6.17	5.17	3.17	+	+	−	−	−	−	−	−
4	5.17	4.17	2.17	+	−	−	−	−	−	−	−
5	4.17	3.17	1.17	+	−	−	−	−	−	−	−
6	3.17	2.17	0.17	−	−	−	−	−	−	−	−
7	2.17	1.17	0.017	−	−	−	−	−	−	−	−
8	1.17	0.17	0.0017	−	−	−	−	−	−	−	−

+: positive result by conventional RT-PCR; −: negative result by conventional RT-PCR.

**Table 4 animals-14-01242-t004:** Stability and detection limit of the Newcastle disease virus using stock virus comparison on Flinders Technology Associates (FTA), using conventional reverse transcription polymerase chain reaction (RT-PCR).

Dilution (log_10_)	Virus Titer (log_10_ ELD_50_/_mL_)	Virus Titer on FTA (log_10_ ELD_50_)	Tested Titer (log_10_ ELD_50_)	Stability on FTA							
				1 wk	2 wk	1 m	2 m	3 m	4 m	5 m	6 m
0	9.83	8.83	6.83	+	+	+	+	+	+	+	−
1	8.83	7.83	5.83	+	+	+	+	+	+	+	−
2	7.83	6.83	4.83	+	+	+	+	+	−	−	−
3	6.83	5.83	3.83	+	+	+	+	+	−	−	−
4	5.83	4.83	2.83	+	−	−	−	−	−	−	−
5	4.83	3.83	1.83	−	−	−	−	−	−	−	−
6	3.83	2.83	0.83	−	−	−	−	−	−	−	−
7	2.83	1.83	0.083	−	−	−	−	−	−	−	−
8	1.83	0.83	0.0083	−	−	−	−	−	−	−	−

+: positive result by conventional RT-PCR; −: negative result by conventional RT-PCR.

**Table 5 animals-14-01242-t005:** Stability and detection limit of the African horse sickness virus using stock virus comparison on Flinders Technology Associates (FTA), using conventional reverse transcription polymerase chain reaction (RT-PCR).

Dilution (log_10_)	Virus Titer (log_10_ PFU/mL)	Virus Titer on FTA (log_10_ PFU)	Tested Titer (log_10_ PFU)	Stability on FTA				
				1 wk	2 wk	1 m	2 m	3 m
0	6.00	5.00	4.01	+	+	−	−	−
1	5.00	4.00	3.01	−	−	−	−	−
2	4.00	3.00	2.01	−	−	−	−	−
3	3.00	2.00	1.01	−	−	−	−	−

+: positive result by conventional RT-PCR; −: negative result by conventional RT-PCR.

## Data Availability

The data presented in this study are available free of charge for any user on request from the corresponding authors.

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
