# Peer review of "Stability and Detection Limit of Avian Influenza, Newcastle Disease Virus, and African Horse Sickness Virus on Flinders Technology Associates Card by Conventional Polymerase Chain Reaction"

_animals, 2024, doi:10.3390/ani14081242_

Round 1

Reviewer 1 Report

Comments and Suggestions for Authors

Q1: Why did you not apply the popular RT-qPCR to quantitatively evaluate the RNA amount captured/fixed on FTA card?  I thought real-time PCR is an excellent tool for your research.

Q2: Here, “the stability of RNA ” can be understood “the integrity of RNA genome”?

Q3: Comparison to AHSV amplicon size of 1167bp, the amplicon sizes for AIV and NDV (of 245bp and 305bp, respectively) are much shorter. So, maybe due to more breaks of the whole genome with the prolonged duration, the longer amplifications (such as 1167 bp for AHSV) are easy to suffer less sensitivity.  Can more different gene’s amplifications solve this problem?  

Q4: Table 1: Why were the primers shown in Talbe 1 newly designed here, instead of using the ones from the references?

Q5:Highly recommend to draw a flow diagram to clarify the whole process. Otherwise, it is hard to understand “the virus titer of stock, virus on FTA card and tested titer”. What is relationship among/between them?

Q6: Result (3.1) : Here, the result’s description is a little simple, and it is hard to understand.

Table 2: Need more information for clarifying the time and dpi.

Q7: How to calculate?  “AHSV used a ratio of 8.0 mm sampling diameter out of 25 mm diameter of FTA card, resulting in a ratio of 1:9.76 (Fig-1).”

Please tell how to calculate it in details in Fig-1. Still a little confused.  

Q8:  Please put the full name for EID50,ELD50 and PFU.

Q9: In abstract (line 29-30),  “Following storage, the target genome was detected using conventional reverse-transcription polymerase chain reaction” and what’s the clear result here?

Q10: The FTA card should be used in the field to verify its efficacy. However, there is no such parts and authors hope to have a look at its application.

Minor mistakes: Line 15 “aimed to evaluated the stability” and some others in the manuscript.

Comments on the Quality of English Language

Excellent! 

Author Response

Thank you very much for your response.

Below, you’ll find answers to all the comments point-by-point.

Q1: Why did you not apply the popular RT-qPCR to quantitatively evaluate the RNA amount captured/fixed on FTA card?  I thought real-time PCR is an excellent tool for your research.

Overall, qPCR offers several advantages over conventional PCR, including the ability to quantify DNA in real-time, increased sensitivity, and reduced turnaround time for results. These features make it a powerful tool in molecular biology research and diagnostic applications. The qPCR is widely used for quantification of gene expression, measurement of viral load, detection of pathogens, SNP genotyping, and other applications requiring precise quantification of DNA. However, conventional PCR is commonly used for applications such as cloning, genotyping and detection of specific DNA sequences, especially, nucleotide sequencing. Hence, the present study aimed to assessed these parameters using conventional PCR.

Q2: Here, “the stability of RNA” can be understood “the integrity of RNA genome”?

Of course, including the capacity of RNA detection by conventional PCR.

Q3: Comparison to AHSV amplicon size of 1167bp, the amplicon sizes for AIV and NDV (of 245bp and 305bp, respectively) are much shorter. So, maybe due to more breaks of the whole genome with the prolonged duration, the longer amplifications (such as 1167 bp for AHSV) are easy to suffer less sensitivity.  Can more different gene’s amplifications solve this problem?

Your assumption is likely correct, and we added this explanation to the discussion section in Line number 395-398. Additionally, we’ll conduct new experiments aiming for shorter amplification, with the hope of maintaining the AHSV genome length longer than in the present study.

Q4: Table 1: Why were the primers shown in Table 1 newly designed here, instead of using the ones from the references?

In present study, all primers were accompanied by references to provide additional context and credibility to the experimental design. Additionally, we made corrections and additions to the references listed in Table 1, ensuring accuracy and completeness of the citation information.

Q5:Highly recommend to draw a flow diagram to clarify the whole process. Otherwise, it is hard to understand “the virus titer of stock, virus on FTA card and tested titer”. What is relationship among/between them?

In our study, we expanded on the definitions of “virus titer on FTA card” and “tested titer”, offering additional clarity for readers. This explanation, along with corresponding details, can be found within Figure 1 and is elaborated upon in the results section (Line number 241-254).

Q6: Result (3.1): Here, the result’s description is a little simple, and it is hard to understand.

Table 2: Need more information for clarifying the time and dpi.

 As per your suggestion, we’ve already addressed the correction outlined in Line number 229-235) including dpi in Table 2.

Q7: How to calculate?  “AHSV used a ratio of 8.0 mm sampling diameter out of 25 mm diameter of FTA card, resulting in a ratio of 1:9.76 (Fig-1).

Please tell how to calculate it in details in Fig-1. Still a little confused. 

As per your suggestion, we have already corrected the error in the result section, specifically in the tested titer calculation (Line number 250-254) and also in Figure 1.

Q8:  Please put the full name for EID50, ELD50 and PFU.

We have already made the correction as per your suggestion in Line number 244, 253 and 281.

Q9: In abstract (line 29-30), “Following storage, the target genome was detected using conventional reverse-transcription polymerase chain reaction” and what’s the clear result here?

We have already made the correction as per your suggestion, and the revised version can be found Line numbers 30-35.

Q10: The FTA card should be used in the field to verify its efficacy. However, there is no such parts and authors hope to have a look at its application.

The correction has been implemented in the conclusion section as per your suggestion, and the revised version can be located from Line numbers 401-423.

Minor mistakes: Line 15 “aimed to evaluated the stability” and some others in the manuscript.

As per your suggestion, the correction has been implemented in the simple summary section, and the revised version can be found from Line numbers 14-23.

Reviewer 2 Report

Comments and Suggestions for Authors

This publication appears to be very similar to previous work by the same group of Authors. 

The previous publication:

"Sensitivity of RNA viral nucleic acid-based detection of avian influenza virus, Newcastle disease virus, and African horse sickness virus on flinders technology associates card using conventional reverse-transcription polymerase chain reaction " with the following identifiers PMCID: PMC9798050 PMID: 359011 Link:https://www.ncbi.nlm.nih.gov/pmc/articles/PMC9798050/ 

What are the advancements or discoveries or improvements being made in the current manuscript? What new data is being presented here?

Comments on the Quality of English Language

Methods section needs to be reviewed by an English editor.

Author Response

This publication appears to be very similar to previous work by the same group of Authors.

The previous publication:

"Sensitivity of RNA viral nucleic acid-based detection of avian influenza virus, Newcastle disease virus, and African horse sickness virus on flinders technology associates card using conventional reverse-transcription polymerase chain reaction " with the following identifiers PMCID: PMC9798050 PMID: 359011 Link:https://www.ncbi.nlm.nih.gov/pmc/articles/PMC9798050/

What are the advancements or discoveries or improvements being made in the current manuscript? What new data is being presented here?

The previous publication elucidated numerous techniques for extracting RNA from the FTA card, including methods to determine the detection limit of viruses on the card. Building upon this groundwork, the present study focuses on assessing the stability of these RNA viruses and the duration required for their detection on the FTA card when stored in a plastic bag at room temperature. These findings not only contribute to our understanding of RNA virus stability but also emphasize the practical utility of FTA cards in virus storage and transportation, thereby facilitating the molecular detection and identification of RNA viral pathogens.

Reviewer 3 Report

Comments and Suggestions for Authors

Thank you for the opportunity to review the article entitled "Stability and detection limit of avian influenza, Newcastle disease virus, and African horse sickness virus on Flinders Technology Associates card by conventional polymerase chain reaction"

The artcle focuses on a crucial part of the effective diagnostics of infections disease, ie the storage and detection sensitivity in Flinder Technology Associates (FTA) membranes particularily on important viral pathogens both in veterinary and human disease control. Heereby I address the following questions and comments:

1. The introduction part is detailed but the manner of too extensive citing in the text could be revised. Some of the citations needs to be compacted and limited to the essential information having the corresponding reference number in the end.

2. More profound comparison with similar studies results should be present for elucidation of method repeatability and specificity issues regarding various pathogens. For example I was suprised not finding papers of Awad et al., 2014 (DOI: 10.1080/03079457.2014.885114),  Linhares et al., 2012 (DOI: 10.1177/1040638711429492), Krambrich et al., 2022 (10.3390/microorganisms10071445) and others.

3. I could not find the reason why Figures 2,3 and 4, as well as Tables 3,4 and 5 are placed in the discussion section but not in the results with the relevant comments coming in the discussion.

4. As this work comes as a logical extension of the authors'paper from 2022 cited  (reference#41) the selection of strains is justified, altough that should be highlighted since H5N1 has much higher clinical significance as an animal and human pathogenic agent with worlwide distribution puting a threat even for Antarctic region dissemination and thus comes as a more logical model for the study.

4. How do authors explain the increased AIV and NDV sensitivity of the test with the prologongation of storage time? How do they expain the much lower stability of AHS viral RNA in FTA

5. Figures of gel electrophoresss might be provided as supplementary material for proof of concept and evidence data.

6. Repetitions such as "Paramyxoviruses, belonging to the family Paramyxoviridae" should be avoided.

With the above mentioned remarks and relevant responses and corrections provided, I could recommend the paper for publishing.

Author Response

Thank you for the opportunity to review the article entitled "Stability and detection limit of avian influenza, Newcastle disease virus, and African horse sickness virus on Flinders Technology Associates card by conventional polymerase chain reaction"

The artcle focuses on a crucial part of the effective diagnostics of infections disease, ie the storage and detection sensitivity in Flinder Technology Associates (FTA) membranes particularily on important viral pathogens both in veterinary and human disease control. Heereby I address the following questions and comments:

  1. The introduction part is detailed but the manner of too extensive citing in the text could be revised. Some of the citations needs to be compacted and limited to the essential information having the corresponding reference number in the end.

We have already implemented the suggested correction in the introduction section, addressing issues identified between line 41-119.

  1. More profound comparison with similar studies results should be present for elucidation of method repeatability and specificity issues regarding various pathogens. For example I was suprised not finding papers of Awad et al., 2014 (DOI: 10.1080/03079457.2014.885114), Linhares et al., 2012 (DOI: 10.1177/1040638711429492), Krambrich et al., 2022 (10.3390/microorganisms10071445) and others.

We have already implemented the suggested correction in the discussion section, specifically addressing issues identified between line 323-340.

  1. I could not find the reason why Figures 2,3 and 4, as well as Tables 3,4 and 5 are placed in the discussion section but not in the results with the relevant comments coming in the discussion.

Following your suggestion, we have relocated all Figure 2-4 and Table 3-5 to the results section. This adjustment enhances the organization of our document, ensuring that all relevant data and visuals are presented together for better clarity and understanding.

  1. As this work comes as a logical extension of the authors'paper from 2022 cited (reference#41) the selection of strains is justified, altough that should be highlighted since H5N1 has much higher clinical significance as an animal and human pathogenic agent with worlwide distribution puting a threat even for Antarctic region dissemination and thus comes as a more logical model for the study.

Thank you very much.

  1. How do authors explain the increased AIV and NDV sensitivity of the test with the prologongation of storage time? How do they expain the much lower stability of AHS viral RNA in FTA

We suggested that the AHSV amplicon size of 1167bp, compared with the amplicon sizes for AIV and NDV (of 245bp and 305bp, respectively) is much shorter. Therefore, longer amplifications (such as 1167 bp for AHSV) may experience lower sensitivity, possibly due to more breaks in the whole genome during prolonged duration. These sentences were included in the discussion section, at line numbers 395-398.

  1. Figures of gel electrophoresss might be provided as supplementary material for proof of concept and evidence data.

We already attached the figure of gel electrophoreses as a supplementary file when we submitted this manuscript.

  1. Repetitions such as "Paramyxoviruses, belonging to the family Paramyxoviridae" should be avoided.

We have already made the correction as per your suggestion, especially regarding the clarification of the introduction section in lines 63-65.

With the above mentioned remarks and relevant responses and corrections provided, I could recommend the paper for publishing.
Thank you very much.

Round 2

Reviewer 1 Report

Comments and Suggestions for Authors

1. In the abstract, it was said that "This study seeks to fill this gap by systematically evaluating.......",however, I did not find what kind of "gap" should be filled. Please make it clear in the abstract. 

2.  The bands shown in the electrophoresis pics were not so clear, especially for the weak bands, please try your best to adjust them.

3. Please double check the references to make sure them matching very well. 

Comments on the Quality of English Language

Minor revision. 

Author Response

Reviewer-1

  1. In the abstract, it was said that "This study seeks to fill this gap by systematically evaluating.......",however, I did not find what kind of "gap" should be filled. Please make it clear in the abstract.

As per your suggestion, we’ve already addressed the correction outlined in Line number 26-29.

  1. The bands shown in the electrophoresis pics were not so clear, especially for the weak bands, please try your best to adjust them.

As per your suggestion, we’ve already addressed the correction all Figure 2-4.

  1. Please double check the references to make sure them matching very well.

Already checked.